# Mechanistic insights into the role of prenyl-binding protein PrBP/δ in membrane dissociation of phosphodiesterase 6

Bilal M. Qureshi[1,2,3,4,8], Andrea Schmidt[1,2], Elmar Behrmann [1,3,5,6], Jörg Bürger[1,3,7], Thorsten Mielke [7], Christian M.T. Spahn[1,3], Martin Heck[1,4] & Patrick Scheerer[1,2]

Isoprenylated proteins are associated with membranes and their inter-compartmental distribution is regulated by solubilization factors, which incorporate lipid moieties in hydrophobic cavities and thereby facilitate free diffusion during trafficking. Here we report the crystal structure of a solubilization factor, the prenyl-binding protein (PrBP/δ), at 1.81 Å resolution in its ligand-free apo-form. Apo-PrBP/δ harbors a preshaped, deep hydrophobic cavity, capacitating apo-PrBP/δ to readily bind its prenylated cargo. To investigate the molecular mechanism of cargo solubilization we analyzed the PrBP/δ-induced membrane dissociation of rod photoreceptor phosphodiesterase (PDE6). The results suggest that PrBP/δ exclusively interacts with the soluble fraction of PDE6. Depletion of soluble species in turn leads to dissociation of membrane-bound PDE6, as both are in equilibrium. This "solubilization by depletion" mechanism of PrBP/δ differs from the extraction of prenylated proteins by the similar folded solubilization factor RhoGDI, which interacts with membrane bound cargo via an N-terminal structural element lacking in PrBP/δ.

[1] Charité – Universitätsmedizin Berlin, corporate member of Freie Universität Berlin, Humboldt-Universität zu Berlin, and Berlin Institute of Health, Charitéplatz 1, D-10117 Berlin, Germany. [2] Charité – Universitätsmedizin Berlin, Institut für Medizinische Physik und Biophysik (CC2), Group Protein X-ray Crystallography and Signal Transduction, Charitéplatz 1, D-10117 Berlin, Germany. [3] Charité – Universitätsmedizin Berlin, Institut für Medizinische Physik und Biophysik (CC2), Group Cryo Electron Microscopy, Charitéplatz 1, D-10117 Berlin, Germany. [4] Charité – Universitätsmedizin Berlin, Institut für Medizinische Physik und Biophysik (CC2), Group Enzyme Kinetics, Charitéplatz 1, D-10117 Berlin, Germany. [5] Research Group Structural Dynamics of Proteins, Center of Advanced European Studies and Research (Caesar), Ludwig-Erhard-Allee 2, D-53175 Bonn, Germany. [6] Institute of Biochemistry—Structural Biochemistry, University of Cologne, Zuelpicher Straße 47, D-50674 Cologne, Germany. [7] UltraStrukturNetzwerk, Max Planck Institute for Molecular Genetics, Ihnestrasse 73, D-14195 Berlin, Germany. [8] Present address: Division of Biological & Environmental Sciences & Engineering, King Abdullah University of Science and Technology (KAUST), 23955-6900 Thuwal, Saudi Arabia. Bilal M. Qureshi and Andrea Schmidt contributed equally to this work. Correspondence and requests for materials should be addressed to P.S. (email: patrick.scheerer@charite.de)

Protein lipidation is one of the major post-translational modifications in proteins. Post-translational lipid-modifications, such as S-prenylation, allows peripheral membrane proteins to associate with membranes[1]. S-prenylation of peripheral membrane proteins occurs exclusively in eukaryotes, where around 2% of proteins (prenylome) undergo irreversible C15-farnesylation or C20-geranylgeranylisation at the cysteine-residue of CAAX box (Cys Aliphatic Aliphatic X). Isoprenylation is followed by proteolytic removal of three C-terminal residues and reversible carboxymethylation of the prenylated Cys at the endoplasmic reticulum[2–4]. Targeting of prenylated proteins to the proper cellular compartment is enabled by several solubilization factors[3–6]. These solubilization factors comprise a hydrophobic cavity, capable of incorporating the lipid-moieties and thereby facilitating free diffusion of prenylated proteins in hydrophilic milieu[7,8]. Intriguingly, several solubilization factors comprise similar 3D-structures despite low primary sequence homology[4,6,7]. They were originally termed GDI or GDI-like (guanine-nucleotide dissociation inhibitors) proteins, due to the discovery of RhoGDI (Ras-homolog GDI) and RabGDI (Ras-related in brain GDI) that interact with prenylated RhoGTPase and RabGTPase proteins, respectively[4,6]. In addition to their role in solubilization and trafficking, both GDIs also inhibit dissociation of GDP in their respective small GTPases[9].

Prenyl-binding protein (PrBP/δ) is a 17 kDa ubiquitous solubilization factor[4,10,11]. In its 3D structure, it is highly similar to other solubilization factors such as RhoGDI or UNC119 (*unco-ordination mutant 119 of C. elegans*) despite a very low sequence identity of 10% and 24%, respectively[4]. PrBP/δ was first discovered in photoreceptor cells as the fourth subunit of cGMP phosphodiesterase 6 (PDE6), hence initially termed PDE6δ[10]. It has also been termed GDI-like protein[6] due to structural and functional similarity to RhoGDI, even though it is not involved in inhibition of guanosine nucleotide dissociation. In photoreceptor cells, PrBP/δ traffics prenylated cargo such as PDE6 and rhodopsin kinase from the site of protein synthesis (inner segment) to a specialized compartment (outer segment)[3,12,13]. However, PrBP/δ is also expressed in the eyeless nematode *C. elegans*[14]. Subsequent investigations have established a role for this ubiquitous protein in various cellular processes apart from photoreceptors due to its interactions with several proteins such as Ras (Rat sarcoma) family of proteins e.g. K-RAS4B and Rheb (Ras-homolog expressed in brain)[15]. Because Ras proteins are involved in several diseases, in particular cancer, prenylation and trafficking of Ras proteins has been identified as a drug target. As targeting farnesyl-transferase showed only limited success in cancer therapy so far[16], PrBP/δ has recently been identified as a potential drug target for cancer therapy and inhibitors of PrBP/δ have been biochemically and structurally characterized[17–19].

An overview of the PrBP/δ mediated trafficking of prenylated proteins from donor to destination membranes is depicted in Fig. 1 (reviewed in refs. [4,20]).

In its simplest form, the PrBP/δ trafficking cycle comprises three steps: (i) solubilization of the prenylated protein (cargo) from its donor membrane by complex formation with PrBP/δ, (ii) targeting to the destination membrane by association of cargo-bound PrBP/δ with a peripherally membrane bound docking protein (e.g., RPGR; Retinitis Pigmentosa GTPase Regulator) that acts as a receptor for PrBP/δ, and (iii) dissociation of the ternary PrBP/δ–cargo–RPGR–complex by interaction of PrBP/δ with displacement factors (e.g., Arl-2-GTP; Arf-like protein) which allows release of the cargo into the destination membrane, as well as recycling of PrBP/δ.

Several complexes of the solubilization factor PrBP/δ have been structurally and functionally characterized. According to these studies, regulated conformational changes of the hydrophobic cavity of PrBP/δ play a key role within its trafficking cycle. In the following, conformations of PrBP/δ with a large ligand cavity, that can accommodate a C15- or C20-isoprene, are termed "open", whereas a conformation with reduced cavity depth, incapable of incorporating the isoprene-moiety, is called "closed" cavity. The cargo-bound form of PrBP/δ (complex of PrBP/δ with, e.g., prenylated Rheb) shows an open conformation of PrBP/δ which accommodates the isoprene moiety[6]. The crystal structure of PrBP/δ in complex with a truncated form of the interacting protein RPGR shows also an open conformation[21]. However, the crystal structure of PrBP/δ with the displacement factor Arl2-GTP (Arf-like protein) shows a drastically reduced cavity size[6,7]. It is widely accepted, that reduction in size of the hydrophobic cavity upon interaction with the displacement factor dislodges the prenylated cargo from PrBP/δ[20]. Accordingly, the displacement factor allows both, association of the cargo with the destination membrane and recycling of apo-form of PrBP/δ[3,6]. Importantly, displacement factor Arl2 interacts with PrBP/δ only in the GTP-bound form[4]; hence, hydrolysis of GTP provides the net-energy input to run through the entire PrBP/δ trafficking cycle.

The focus of this work is to characterize the initial step of the PrBP/δ trafficking cycle, i.e., the PrBP/δ induced dissociation of prenylated proteins from the donor membrane (Fig. 1). Here we present a high-resolution crystal structure of the isolated ligand-free apo-form of PrBP/δ (apo-PrBP/δ), a low-resolution electron microscopy structure of PrBP/δ bound to cGMP phosphodiesterase 6 (PDE6), and a comprehensive biochemical analysis of PrBP/δ induced membrane dissociation of PDE6. Based on these structural and functional data we delineate a conclusive model for solubilization of peripheral membrane proteins by PrBP/δ.

## Results

**Overall fold of the prenyl-binding protein PrBP/δ.** The crystal structure for the N-terminally His-tagged apo-PrBP/δ monomer was solved at a resolution of 1.81 Å (for data collection, structure determination and refinement statistics, see Methods and Table 1). In the final 2m*Fo*-D*Fc* electron density map all amino acids with the exception of a flexible loop (A111 - A117) and additional four amino acids of the N-terminal tag are clearly visible. The cargo-free structure of apo-PrBP/δ monomer obtained here shows the typical immunoglobulin-like fold: a β-sandwich comprising two anti-parallel β-sheets packing against each other (Fig. 2a). Intriguingly, the interior of the β-sandwich harbors a deep hydrophobic cavity, which provides optimal environment for a prenyl-moiety. This open accessible cavity is ~19 Å deep and ~10 Å wide; therefore, C15-farnesylated or C20-geranylgeranylated cargos may readily bind the PrBP/δ. As mentioned in the introduction, two basic conformations of PrBP/δ exist, "open" with a deep cavity and "closed" with a reduced cavity size. As we compare and discuss in greater detail in the following paragraph, we obtained the open conformation of PrBP/δ. In the apo-state a preformed cavity may allow an unspecific exchange of small molecules. We found in the cavity some, but not well-defined, electron density which indicate the presence of a flexible, small and polar molecule. Therefore we assigned this electron density as water molecules, which are located in H-bond distance to Arg61 (Supplementary Fig. 1). Defining the entrance of the hydrophobic cavity as being located at the upper front side, the apo-form of PrBP/δ shows a negatively charged front with a positive surface charge at the upper rim of PrBP/δ and smaller negatively charged surface area on its back side (Supplementary Fig. 2).

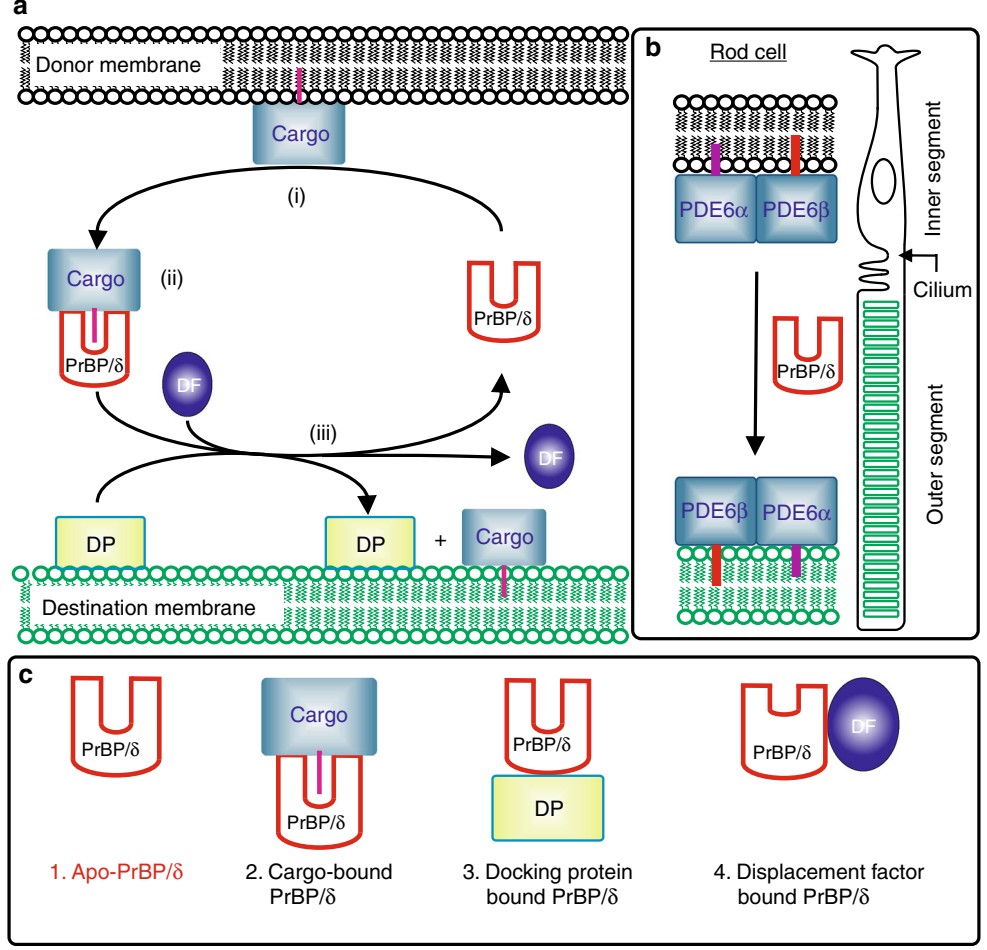

**Fig. 1** Overview of interactions of the solubilization factor PrBP/δ. **a** Schematic model of the PrBP/δ trafficking cycle. The transport of prenylated proteins (cargo) is initiated upon (i) dissociation of the cargo from the donor membrane by interaction with PrBP/δ (red). (ii) Cargo-bound PrBP/δ then diffuses through the cytoplasm, as the hydrophobic isoprene moiety of the cargo is buried in a cavity of PrBP/δ. Subsequent targeting of cargo-bound PrBP/δ to the destination compartment is achieved by interaction of PrBP/δ with docking proteins (DP) that are peripherically bound to the destination membrane. Additional GTP/GDP-binding proteins (dissociation factors, DF) are required to dissociate the cargo from PrBP/δ which in turn allows (iii) release of the cargo into the membrane[6]. Eventually, the trafficking cycle closes when apo-PrBP/δ diffuses back to the donor compartment. **b** Schematic representation of a rod photoreceptor cell. The rod cell consists of an inner segment (with the common cell compartments) and, connected by a cilium, an outer segment, which contains a stack of disc membranes and the signaling proteins. Binding of PrBP/δ to newly synthesized, prenylated signaling proteins such as PDE6 allows their transport to the outer segment. **c** Comparison of selected crystal structures of PrBP/δ. Beside the crystal structure of (1) apo-PrBP/δ presented in this study, several PrBP/δ-complexes are structurally characterized, including (2) cargo-bound PrBP/δ (e.g., farnesylated Rheb (PDB entry 5T3G); farnesylated INPP5E-peptide (PDB entry 5F2U); farnesylated K-RAS4B (PDB entry 5TAR); geranylgeranylated PDE6C-peptide (PDB entry 5E8F)), (3) PrBP/δ bound to docking protein RPGR (PDB entry 4JHP) and (4) PrBP/δ bound to the displacement factor Arl2 (PDB entry 1KSG). Crystal structures of (1) and (2) can be assigned to the trafficking cycle (**a**), whereas (3) and (4) are part of the dissociation step of the cargo into the destination membrane

**Comparison of various functional states of PrBP/δ.** The obtained structure of apo-PrBP/δ (PDB entry 5T4X) with its preshaped cavity in open conformation was compared with examples of known PrBP/δ conformations from earlier published co-crystal structures (Fig. 2).

For cargo-bound PrBP/δ several complexes with prenylated proteins or peptides are known[22], PrBP/δ in complex with farnesylated and methylated K-RAS4B at 2.0 and 1.9 Å resolution[23], PrBP/δ in complex with farnesylated INPP5E peptide at 1.85 Å resolution[24] and PrBP/δ in complex with farnesylated RHEB at 1.7 Å resolution[6].

The Rheb-bound PrBP/δ (PDB entry 3T5G, at 1.7 Å resolution[6]) depicts the crystal structure of PrBP/δ with its cargo, the farnesylated Rheb, and shows an open conformation. Binding of the C15-farnesyl-moiety of Rheb hardly influences size of the cavity, but induces side chain rearrangements within the cavity to allow optimal incorporation of the isoprene (Fig. 2;

Supplementary Fig. 3b). Structural similarities of apo- and cargo-bound form are reflected by a low root-mean-square deviation (RMSD) of 0.96 Å of all atoms. Nevertheless, the main differences between the two open conformations are located at the cavity entrance that forms an interface with the cargo. The flexible loop, localized upside of the cavity and invisible in the apo-PrBP/δ, becomes structurally defined due to the interaction with the cargo protein (Fig. 2b). The isoprene of the cargo fits well and defines the cavity; especially Trp90 slightly shifts (1.3 Å) in the Rheb bound PrBP/δ complex towards the cargo and contributes to the binding of the farnesyl-moiety (Supplementary Fig. 3b). In the K-RAS4B bound PrBP/δ complex Trp90 shows a different rotameric conformation as in Rheb-bound or apo-form. Based on the substitution Trp90Ala, Dharmaiah and coworkers describe that Trp90 shows no influence on K-RAS4B binding[23]. However, in contrast to K-RAS4B-PrBP/δ complex where Trp90 undergoes an outward rotation that may allow more extensive

**Table 1 Data collection and refinement statistics**

|  | Apo-PDE/δ[a] (PDB entry 5T4X) |
|---|---|
| Data collection[a] | ESRF, ID29 |
| (wavelength) | $\lambda = 0.966$ Å |
| Space group | $P2_12_12_1$ |
| Cell dimensions |  |
| $a, b, c$ (Å) | 27.66 56.53 89.10 |
| $\alpha, \beta, \gamma$ (°) | 90.0, 90.0, 90.0 |
| Resolution (Å) | 47.74–1.81 (1.91–1.81)[b] |
| $R_{merge}$ | 0.09 (0.68) |
| $R_{pim}$ | 0.05 (0.30) |
| $<I/\sigma(I)>$ | 10.4 (2.3) |
| $CC1/2$ | 99.8 (79.3) |
| Completeness (%) | 99.6 (99.8) |
| Redundancy | 5.3 (5.5) |
| Wilson B factor (Å$^2$) | 21.8 |
| Refinement |  |
| Resolution (Å) | 47.74–1.81 |
| No. Reflections | 126551 |
| $R_{work}/R_{free}$ (%) | 18.3/21.4 |
| Overall B Factor ((all atoms; Å$^2$) | 27.30 |
| B-factors |  |
| Protein | 26.70 |
| Ligand/ion |  |
| Water | 34.70 |
| No. atoms/residues (1 monomer per asymmetric unit) |  |
| Protein (apo-PrBP/δ) | 1227/148 |
| Water | 85 |
| R.m.s[c] deviations |  |
| Bond lengths (Å) | 0.008 |
| Bond angles (°) | 1.1 |
| Ramachandran plot[d] |  |
| % favored | 98.0 |
| allowed | 2.0 |
| outlier | 0.0 |

[a] One crystal was used
[b] highest resolution shell is shown in parenthesis
[c] R.m.s. root mean square
[d] Ramachandran plot created by RAMPAGE using the Richardsons' data

interaction with the cargo[23], the Rheb-PrBP/δ complex reveals for Trp90 a shift into the cavity towards the isoprene moiety (Fig. 2; Supplementary Fig. 3c). In the Rheb–PrBP/δ complex Trp90 participates at interaction interfaces to isoprene as well as to the C-terminus of cargo. In apo-form crystal structure Glu88, which has been shown to be involved in cargo binding[23], exists in two rotameric conformations. The high flexibility of Glu88 in the apo-form on the one hand and its defined, stabilized conformations in the cargo-bound structures on the other hand, seem to reveal that Glu88 is required to adapt to various conformations depending upon bound-cargo. Interestingly, Glu88 and Trp90 are hydrogen bonded in the majority of the crystal structures. The incorporation of a C20-geranylgeranyl-moiety into PrBP/δ requires an additional adjustment. The residue Phe133, which is part of the bottom of the hydrophobic cavity, undergoes a downwards shift to accommodate the longer isoprene[23]. Altogether, the hydrophobic cavity in PrBP/δ in apo-form is preshaped and ready to incorporate the isoprene moiety of the cargo, but specifically adapts upon cargo binding.

Next we compared PrBP/δ in complex with the membrane anchoring factor RPGR, (PDB entry 4JHP)[21] (Fig. 2a, b). The localization of RPGR in the destination membrane allows membrane-targeting of the cargo. Among the PrBP/δ conformations, this complex-form shows highest similarity with the apo-form with a RMSD of 0.72 Å of all atoms. Furthermore, PrBP/δ also shows an open conformation with an empty cavity in

complex with RPGR. Differences in the entrance of the cavity are limited to the flexible loop, which result in a different surface charge of PrBP/δ (Fig. 2a). A closer look into the hydrophobic cavity shows that Phe133 of the RPGR-bound complex sticks into the cavity of the apo-form, which again shows the flexibility of Phe133 and thereby underlines its role in adaptation mechanism of the cavity upon cargo binding (Fig. 2c). After RPGR binding, PrBP/δ fits by side chain rearrangements (e.g., Lys29, Gln33, Thr35, Arg48, and Lys51) to the new binding partner, while the overall secondary structure of PrBP/δ remains the same, which is necessary to keep the cargo bound. Finally, we compared the open conformation of apo-PrBP/δ with the closed conformation of PrBP/δ in complex with its displacement-factor (Arl2) (PDB entry 1KSG)[7] (Fig. 2a, b). These two conformations reveal dramatic differences with an RMSD of 2.36 Å of all atoms. The main difference lies in the depth of the cavities which is in the open apo-PrBP/δ approximately 18 Å whereas in closed conformation only ~6 Å, excluding binding of any isoprenes in the closed state. As described by Ismail and coworkers[6] closure of PrBP/δ is due to a narrowing of the two β-sheets of PrBP/δ, which form the hydrophobic cavity, structurally based on a β-sheet extension over the two molecules of PrBP/δ and its release factor Arl2 (Fig. 3b). Such a β-sheet extension is also formed in apo-PrBP/δ crystal structure in which two crystallographic symmetry related monomers interact via a β-strand 6 interface (Fig. 3a). By comparing the apo-PrBP/δ with this PrBP/δ-Arl2 complex we assume that in addition to the β-strand extension a key residue for the closure mechanism in PrBP/δ is Phe94 (Fig. 3). The rotameric conformation of Phe94 in the apo-PrBP/δ crystal structure would disturb a potential binding to Arl2 due to the conformation of Phe50 in Arl2 (Fig. 3c*). Triggered by binding of Arl2 to PrBP/δ, Phe50 can induce a rotameric shift of Phe94 and therefore seems to start a cascade of several side chain changes including amino acid Trp105, Trp90, Glu88 and Arg61 (Fig. 3c). In the closed conformation of PrBP/δ Arg61 (green sticks in Fig. 3c) is able to build up a new hydrogen bonding network with Glu88, Trp90 (residues involved in cargo-binding or cavity adaptation) and Gln78 that stabilize the narrowing of the β-sheets (Figs. 2c and 3c).

To exclude that the β-sheet extension in apo-PrBP/δ (with its symmetry related molecule) does not induce an artificial conformation for the β-strand 6, we compared apo-PrBP/δ also with PrBP/δ-Rheb and -RPGR complex crystal structures, where no interface interaction at this strand are found. From these comparisons it is plausible that no structural changes at β-strand 6 occur in the crystallographic interface. Thus the β-sheet extension alone is not able to induce closing of the cavity of PrBP/δ. Interestingly, similar to the aromatic Phe50 in Arl2 we found in the symmetry related molecule of apo-PrBP/δ a histidine residue, whose side chain could also introduce a clash but points as another rotamer to the opposite direction. This underlines the model, that a correct positioning of the β2 strand of Arl2 and therefore of Phe50 is a prerequisite to close the cavity. The correct positioning may be realized by additional loop-loop interactions of PrBP/δ and Arl2 that are absent in apo-PrBP/δ alone. Therefore, these results extend the observations of earlier studies[6].

Besides Arl2, another displacement factor for PrBP/δ is known, namely Arl3. Although Arl2 and Arl3 act on different cargos, the mechanism of cargo release via closure of the PrBP/δ cavity is similar. Arl2 is identified as displacement factor for low affinity cargo, whereas Arl3 displaces high affinity complex cargo of PrBP/δ in cilia[22]. For our investigations we used the rod phosphodiesterase 6 (PDE6), which requires PrBP/δ mediated trafficking from the inner segment of the rod cell via the cilium to the rod outer segment. Since PDE6 is a high affinity PrBP/δ-cargo

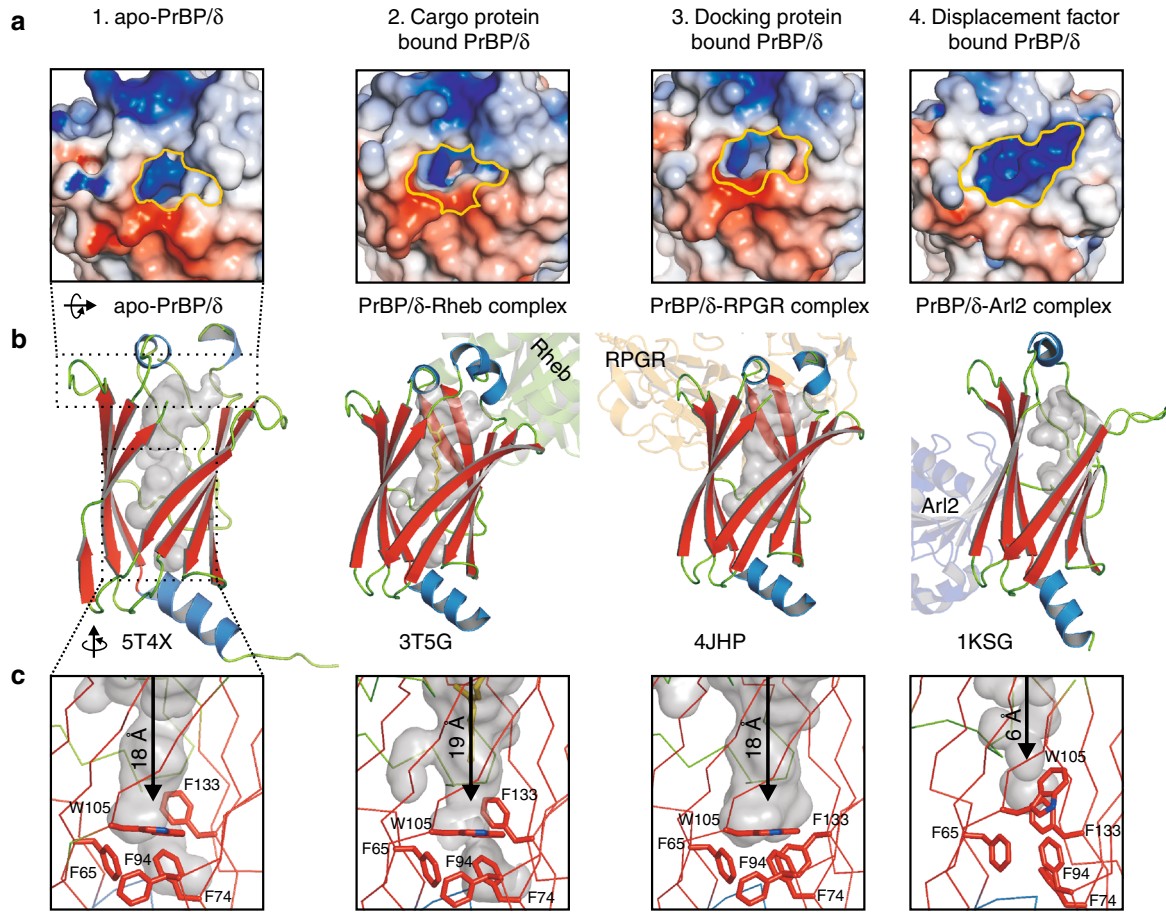

**Fig. 2** Prenyl-binding protein (PrBP/δ) crystal structures in various conformations. **a** Electrostatic surface representation of prenyl-binding protein (PrBP/δ) crystal structures. Electrostatic surface potentials were calculated using the program APBS with the non-linear Poisson–Boltzmann equation and contoured at ±5 kT e$^{-1}$ [78]. Negatively and positively charged surface areas are colored in red and blue, respectively. To elucidate the entrance of the cavity a yellow edge was introduced manually. The structured loop above the entrance in the PrBP/δ-Rheb-bound complex was removed in front for clarity. The cavity entrance of the three open conformations of PrBP/δ (PDB entries 5T4X, 3T5G, 4JHP) are more similar compared to the entry site of the displacement factor bound PrBP/δ in the closed conformation (PDB entry 1KSG). The apparent larger cavity entrance in 1KSG is a result of the increased flexibility in the flexible loop above the entrance. Moreover, the closure of the cavity results in an electrostatic surface potential rearrangement which may facilitate the release of the cargo. **b** Overall fold of PrBP/δ in cartoon representation with the hydrophobic cavity depicted as gray surface. The C15-farnesyl-moiety in the cargo-bound form is shown as yellow sticks. Binding partners of PrBP/δ are depicted in cartoon representation for clarifying the different additional protein interfaces in contrast to apo-PrBP/δ. **c** Close-up view of the PrBP/δ binding cavities. The PrBP/δ binding cavities are shown as ribbon representations and surfaces (gray). Key aromatic residues Phe65, Phe74, Phe94, Trp105, and Phe133, which define the bottom of the hydrophobic pockets for all selected crystal structure conformations, are depicted as stick representation. In particular, Phe94 and Trp105 of PrBP/δ indicate a rotation of their side chains after complex formation with the displacement factor (PDB entry 1KSG) and thereby a significant reduction of the cavity. Therefore, the PrBP/δ–Arl2 complex is incapable of incorporating any C15- or C20-isoprenyl-moieties

it is assumed that Arl3 enables the displacement of PDE6 into the destination membrane.

**Characterization of the PDE6–PrBP/δ complex.** We next investigated the interaction of PrBP/δ with the rod photoreceptor cGMP phosphodiesterase 6 (PDE6). PDE6 is transported from the rod inner segment to the disc membranes of outer segment by PrBP/δ (see Fig. 1b), where it co-localizes with rhodopsin, transducin etc, as also advocated by PrBP/δ mouse knock-out studies[3,13,25]. Since both catalytic PDE6 subunits are isoprenylated, it is expected that two molecules of PrBP/δ bind one PDE6-holoenzyme[26–28]. Despite two isoprene-moieties, contradicting reports about the stoichiometry of the complex are available[12,29,30]. Therefore, we applied analytical size-exclusion chromatography to verify the 1:2 stoichiometry of the PDE6-PrBP/δ complex in solution. The experiment (Fig. 4a) shows that PDE6 and apo-PrBP/δ co-migrate, i.e., their peaks merge in a

bigger peak that is shifted towards higher molecular weight. SDS-PAGE analysis of fractions from size-exclusion chromatography confirms that both proteins co-eluted (Supplementary Fig. 4). Moreover, twofold molar excess of apo-PrBP/δ completely bound to PDE6, while additional PrBP/δ (e.g., at a ratio of 1:3 PDE6: apo-PrBP/δ) eluted as a separate peak (Fig. 4a). This confirms that the C15-farnesyl moiety of PDE6α as well as C20-geranylgeranyl moiety of PDE6β binds each an apo-PrBP/δ molecule and no additional factors such as membranes are required to promote the interaction of PDE6 and PrBP/δ. In order to further verify the complex stoichiometry we performed low-resolution electron microscopy of the soluble PDE6–PrBP/δ complex. Class-averages were generated from PDE6 and PDE6–PrBP/δ samples. Averages of all classes (Fig. 4b) show the bell-shaped PDE6 both with and without PrBP/δ molecule, similar to earlier structures of PDE6[29,31–33]. Importantly, representative classes for PDE6 sample show the typical PDE6 density

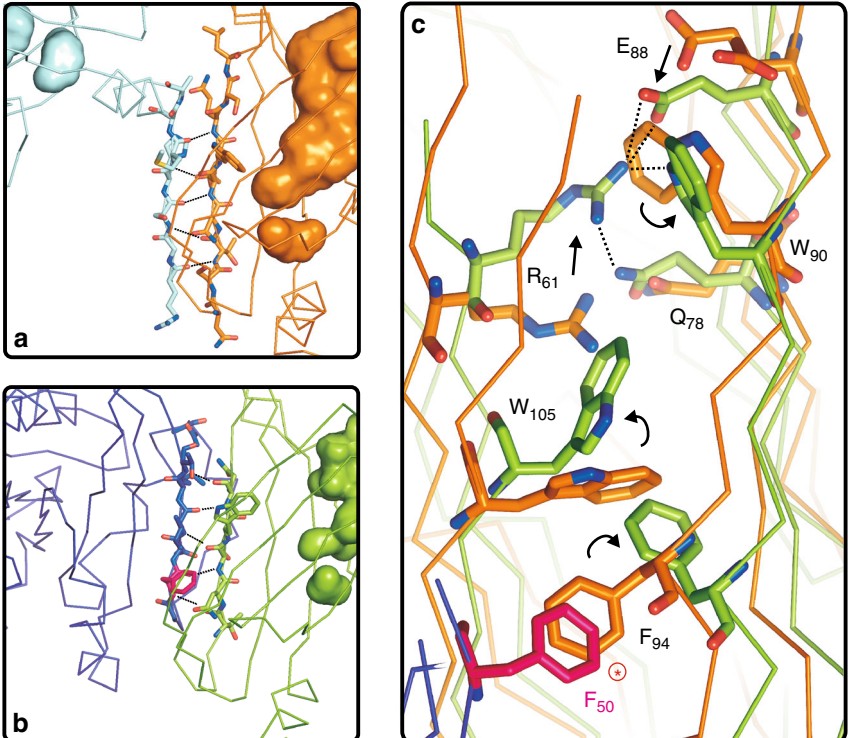

**Fig. 3** Comparison of open and closed conformation of PrBP/δ. **a** Apo-PrBP/δ in "open" conformation. Apo-PrBP/δ in open conformation is shown as ribbon representation with a cavity-surface (cyan/orange). A β-sheet extension is formed due to β-strand 6 interaction based on the crystallographic interface of apo-PrBP/δ (orange) with N-terminus of the symmetry related apo-PrBP/δ molecule (cyan). **b** PrBP/δ in "closed" conformation. In the crystal structure complex of PrBP/δ with its displacement factor Arl2 a very similar β-sheet extension is formed in which the β-strand 6 of PrBP/δ (green) interacts with β2 strand of Arl2 (purple). Phe50 of Arl2 (highlighted in pink) (**c**) seems to induce a rotameric shift of Phe94 in PrBP/δ. For both apo-PrBP/δ ("open" conformation) and Arl2-bound PrBP/δ ("closed" conformation) this extension of the β-sheet with β6 is observed. **c** Superposition of the apo-PrBP/δ in "open" (orange) and PrBP/δ-Arl2-bound in closed (green) conformation. Closure of the PrBP/δ cavity is apparently induced by the Phe50 of Arl2 that might induce a cascade of side chain shifts of several residues such as Phe94, Trp105 and Arg61. These shifts allow formation of a new hydrogen bonding network including the residues Arg61, Gln78, Glu88, and Trp90 in the "close" conformation that are additionally responsible for the size reduction of the cavity. In contrast the distance between Arg61 and Glu88 is ~10 Å in the "open" conformation of apo-PrBP/δ

but only a fraction (~5% of all particles) contain one or two additional densities protruding from the catalytic domain, likely corresponding to the ~17 kDa PrBP/δ molecule. These 5% complexes likely arise from purification of native PDE6 that contain some residual PrBP/δ molecules. Representative classes of PDE6–PrBP/δ sample show predominantly (~78% of particles) one (PDE6–PrBP/δ) or two (PDE6–PrBP/δ₂) additional densities corresponding to PrBP/δ. Therefore, electron microscopy corroborates our finding that each PDE6 molecule can bind up to two PrBP/δ molecules. Furthermore, various orientations of PrBP/δ suggest that the C-terminal of PDE6 is flexible, explaining lack of any protruding densities for average images of classes. Such a flexible C-terminal may also have a role in membrane-binding and orientation of PDE6 molecules on native membranes.

**Affinity of PrBP/δ for PDE6.** The affinity of PDE6 for native disc membranes is in the micro molar range[34]. At high PDE6 concentrations as it prevails in native rod outer segments (~30 μM), most of the protein is thus associated to the disc membranes. At lower overall concentrations, however, a significant fraction of PDE6 remains in solution. We used a centrifugal pull-down assay to quantify the fraction of soluble (PDE6ₛ~33%) and membrane bound PDE6 (PDE6ₘ~67%) under the conditions of our experiments and in the absence of PrBP/δ (Supplementary Fig. 5). Notably, membrane associated PDE6 is activated much more efficiently by the active (i.e., GTP- or GTPγS-bound) α-subunit of

transducin (Gα*) as compared to soluble PDE6[35–38]. We thus measured the Gα*-stimulated activity of PDE6 to monitor the PrBP/δ-induced dissociation of PDE6 from the disc membranes. As seen in Fig. 4c, PDE6 activity decreases with increasing concentration of PrBP/δ to a level comparable to the PDE6 activity measured in the absence of membranes. Because binding of PrBP/δ to PDE6ₛ does not affect the enzymatic PDE6 activity[11,39] the decrease of the Gα*-induced PDE6 activity directly reflects the PrBP/δ induced dissociation of PDE6ₘ from the membranes.

Interaction of PrBP/δ with PDE6 was quantified using the reaction model depicted in Fig. 5 (right panel). The model assumes that (i) PrBP/δ binds exclusively to PDE6ₛ and (ii) PDE6ₛ comprises two independent binding sites for PrBP/δ. Accordingly, the progressive depletion of membrane-associated PDE6 is due to a shift of the coupled equilibria upon successive formation of PDE6–PrBP/δ and PDE6-(PrBP/δ)₂ in solution. Because PDE6 contains two different isoprene-moieties (i.e., C15-farnesyl at PDE6α and C20-geranylgeranyl at PDE6β) it is likely that the affinities of PrBP/δ for the two binding sites on PDE6 are not equal. Indeed, the data points of the titration curve shown in Fig. 4c (activity of PDE6 as a function of PrBP/δ) are well fitted with two $K_D$ values, namely $K_2 = 0.04 \pm 0.01$ μM and $K_3 = 0.7 \pm 0.9$ μM, respectively. The ~20-fold relative difference of the two values is in good agreement with previous results obtained with PrBP/δ and C15-farnesyl-cysteine ($K_D = 0.7$ μM) and C20-geranylgeranyl-cysteine ($K_D = 19$ μM), respectively[12]. We may thus assign the higher affinity site to the PDE6α subunit and the lower affinity site

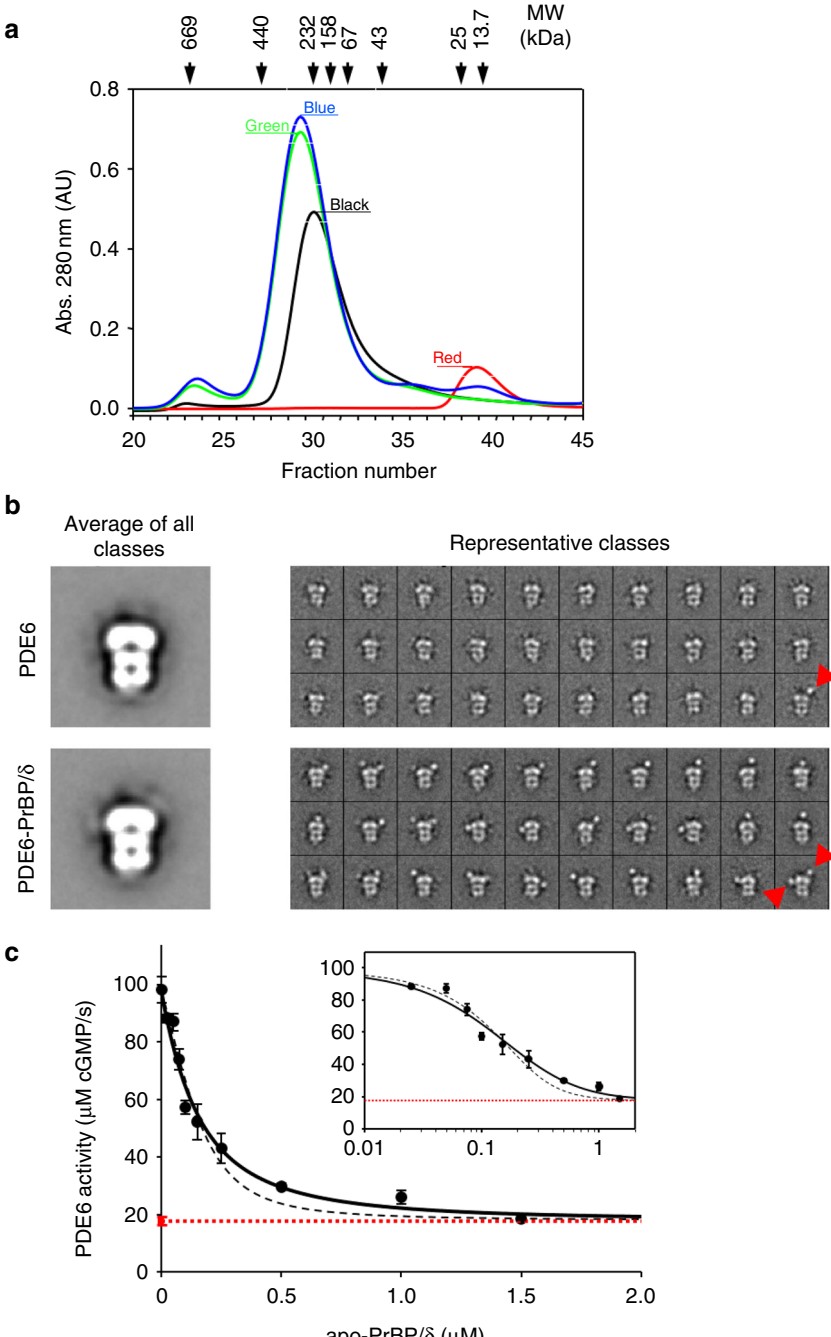

**Fig. 4** Interaction of PrBP/δ with the PDE6. **a** Size-exclusion chromatography of PDE6–PrBP/δ complex. Analytical size-exclusion chromatography confirms the stoichiometry of the complex as PDE6–PrBP/$\delta_2$. Runs for PDE6 only (black; molecular weight (MW) of ~217 kDa) and PrBP/δ only (red; MW of ~17 kDa). Gel filtration reveals complex formation and peak shift for PDE6–PrBP/δ at ratios of 1:2 (green) and 1:3 (blue). Excess PrBP/δ at the ratio of 1:3 (blue) elutes as a separate peak at MW of ~17 kDa (Supplementary Fig. 4). **b** Electron microscopy of the PDE6–PrBP/δ complex. Averages of all classes for PDE6 and PDE6–PrBP/δ complexes obtained by electron microscopy show a bell-shaped PDE6 molecule, where the densities at the top represent the more massive catalytic domains. Representative class-averages of PDE6 sample show one or two additional protruding densities for around 5% particles, which likely represents endogenous PrBP/δ (~17 kDa) of the native sample. Representative class-averages of PDE6–PrBP/δ sample show one or two additional protruding densities for around 78% particles, which represents PDE-bound PrBP/δ. Additional protruding densities are highlighted by red arrows in the lower right box of each panel of classes. **c** Activation assay of PDE6 as a monitor of PrBP/δ induced solubilization of PDE6. Gα*-induced activity of PDE6 (50 nM; circles) was measured as a function of PrBP/δ concentration in the presence of isolated disc membranes (10 μM) and Gα* (1 μM). Increasing concentration of PrBP/δ progressively decreases PDE6 activity till it was equal to the PDE6 activity measured in solution (dotted red line). Data points were fitted to a model with two different PrBP/δ binding sites on PDE6 (solid black line; $K_1 = 0.04 \pm 0.01\,\mu M$ and $K_2 = 0.7 \pm 0.9\,\mu M$; $R^2 = 0.996$) or with two identical binding sites (dashed gray line; $K = 0.07 \pm 0.01\,\mu M$; $R^2 = 0.994$). Inset: semi-logarithmic replot of the data. All data points are replicates of three measurements depicted with standard error

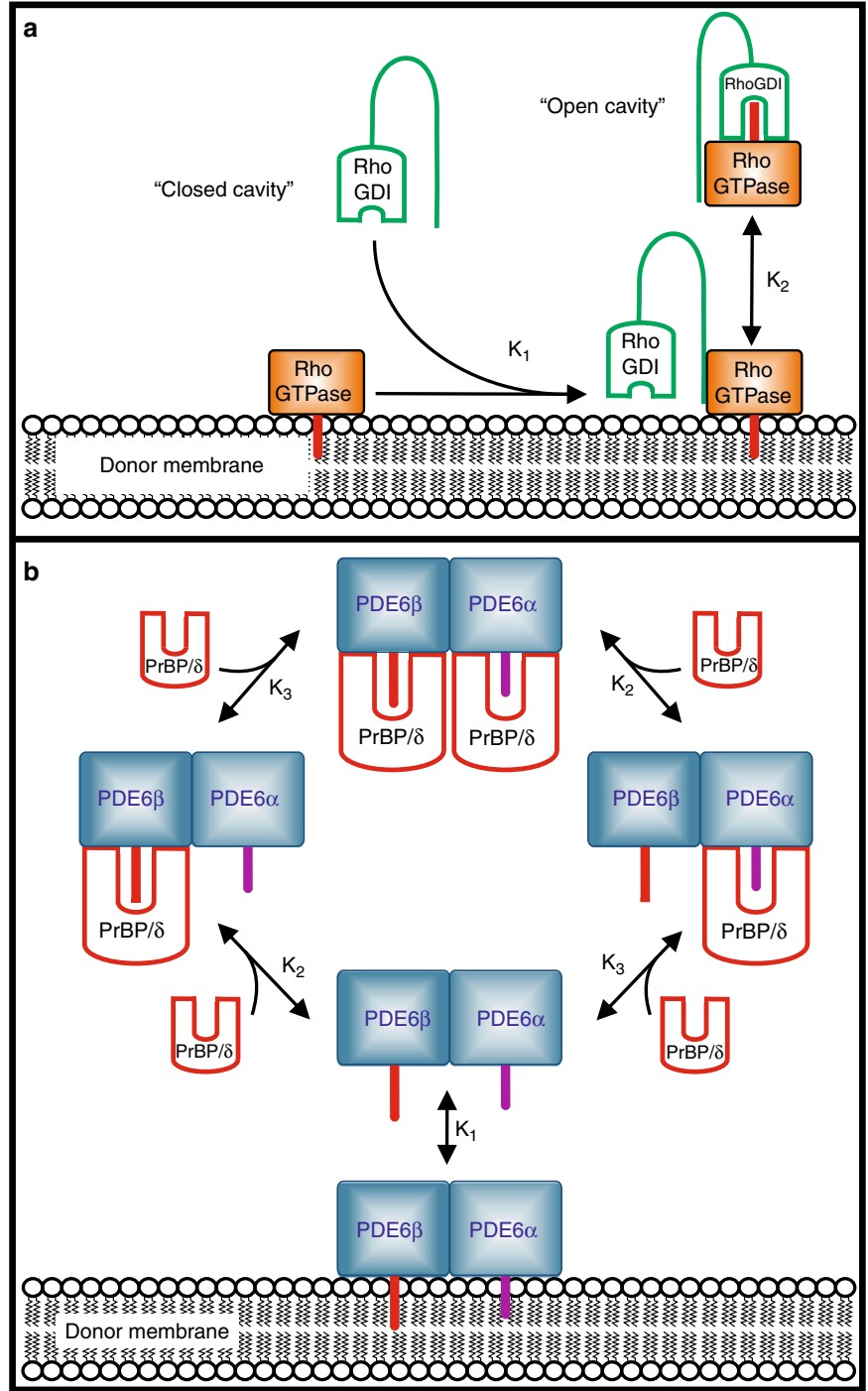

**Fig. 5** Functional comparison of PrBP/δ and RhoGDI. **a** Schematic model of RhoGDI-induced solubilization of RhoGTPase. The cavity of apo-RhoGDI (PDB entry 1DOA)[41] is in the closed conformation. Interaction of the N-terminal "regulatory arm" of RhoGDI with its binding partner on the membrane induces opening of the cavity and allows translocation of the isoprene moiety (symbolized by the red stick) of the RhoGTPase from the membrane into the cavity of RhoGDI and thereby solubilization of the RhoGTPase. **b** Schematic model PrBP/δ-induced solubilization of PDE6. Apo-PrBP/δ harbors a preshaped deep hydrophobic cavity but lacks an N-terminal domain that could function like the regulatory arm of RhoGDI. The doubly-prenylated PDE6 exists in equilibrium of membrane-associated and soluble species. Dissociation of membrane-associated PDE6 involves sequestering of soluble fraction of PDE6 by two copies of PrBP/δ. Depletion of soluble species of PDE6 eventually leads to complete dissociation of the membrane-associated fraction ("dissociation by depletion mechanism"; see text for details). Farnesyl- (purple color) and geranylgeranyl moieties (red color) are shown as sticks

to the PDE6β subunit. Fitting of the titration data with a single type of binding sites on PDE6 for PrBP/δ yields $K = 0.07 \pm 0.01\,\mu M$ (see Fig. 4c). Importantly, the proposed mechanism of membrane dissociation (see next paragraph) does not depend on the relative affinity of PrBP/δ for the two sites on PDE6.

## Discussion

Despite vast studies on PrBP/δ, little is known about the molecular mechanism of the extraction of prenylated cargo-proteins from the membrane[40]. For the homologous solubilization factor RhoGDI a two-step mechanism of membrane extraction of

prenylated proteins is suggested and briefly reviewed here for comparison. RhoGDI, as discussed earlier, harbors a similar 3D-fold and hydrophobic cavity as PrBP/δ (Fig. 5), despite low sequence identity. In contrast to the preshaped open cavity of isolated PrBP/δ, however, the cavity of isolated RhoGDI is in the closed conformation[41]. First step of interaction with prenylated binding partners such as the RhoGTPases cdc42 (cell division cycle 42)[41–44] or RhoA[45] is a low affinity interaction on the membranes via an N-terminal "regulatory arm" (Fig. 5c). A moderate affinity complex is then formed on the membranes, as the entry site of the closed cavity becomes proximal to the membrane-inserted isoprene due to additional electrostatic interactions between the cavity entrance of RhoGDI and C-terminal of the cargo protein. This brings the prenyl-moiety of the cargo proximal to the cavity entrance site. Subsequent translocation of the prenyl-moiety from membrane into RhoGDI extends and deepens the bottom of hydrophobic cavity, rendering it in an open conformation[44,45]. Taken together, cargo binding occurs via induced fit mechanism on the membrane and the eventual formation of the soluble, high affinity (~pM) complex provides the driving force for the solubilization, which was thus termed "binding energy switch"[45].

Several lines of evidence suggest that the mechanism of cargo extraction from membranes differs between RhoGDI and PrBP/δ. Recently, Schmick and colleagues[40] have performed cell-based in vivo investigations on PrBP/δ mediated solubilizaion of KRas from endo-membranes and concluded that binding of KRas to PrBP/δ is a passive sequestration of spontaneously dissociated KRas. Consistently, there is no evidence for formation of a membrane-bound complex of PrBP/δ with cargo-proteins[15]. From a structural point of view PrBP/δ lacks an N-terminal domain that could function like the regulatory arm of RhoGDI. It is thus unlikely that PrBP/δ can interact with its prenylated binding partner prior to its dissociation from the membrane. In our crystal structure, isolated PrBP/δ (apo-PrBP/δ) exhibits an open conformation, i.e., harbors a deep, hydrophobic cavity even without accommodating a prenyl-moiety. Taken together, these findings strongly support a "dissociation by depletion" mechanism for PrBP/δ (Fig. 5). According to this model, the affinity of the prenylated cargo such as PDE6 for the membrane is low enough to establish an equilibrium between a membrane associated and soluble fraction. The preshaped, open cavity allows PrBP/δ to readily interact with the soluble fraction of PDE6. The initial step in membrane extraction is thus an interaction between the cargo and PrBP/δ in solution rather than a transient complex formation on the membrane. Hence, binding of cargo by PrBP/δ occurs via conformational selection type of mechanism. Complex formation in solution depletes soluble PDE6 and thereby induces further dissociation of the membrane-associated fraction until eventually the cargo is completely extracted from the membrane.

The physiological role of PrBP/δ is to traffic PDE6 from inner to the outer segments of the photoreceptors via the connecting cilium[3,25]. PrBP/δ is predominantly localized in the inner segment, where it forms the soluble complex with PDE6, whereas PDE6 is highly concentrated in the outer segment. The high membrane concentration in the outer segment (which contains a stack of about 1000 disks) functions as a sink for PDE6, which is supplied from the inner segment. Furthermore, the continuous distal migration of mature disks and formation of new disks at the ciliary plasmalemma creates binding sites for new PDE6 molecules. In general, the sole function of PrBP/δ and also UNC119 (also termed RG4; Retinal Gene 4) is to facilitate the free diffusion and thus trafficking of prenylated proteins. In the case of the rod cell, unidirectionality is provided by law of mass action (e.g., high concentration of the acceptor membrane), and is enhanced by GTPases[46] such as Arl3 involved in release of the cargo.

Consistently, PrBP/δ knock-out[5] or Arl3 knock-out[47] leads to only partial loss of transport. In contrast to the trafficking protein PrBP/δ, the physiological role of RhoGDI is not only to detach RhoGTPases from membranes but also to "keep them inactive" (GDI-function; guanine dissociation inhibition). In general, solubilization factors such as RhoGDI regulate RhoGTPases in nucleotide synchronized fashion along with GEFs (guanine nucleotide exchange factors) and GAPs (GTPase activating proteins) rather than to traffic them between compartments[48,49]. Direct extraction by formation of membrane-associated complex may therefore offer a more efficient and reliable regulation, which is evident from the 1000-fold decrease in affinity of RhoGDI for RhoA upon exchange of nucleotide within the prenylated cargo[45]. Hence, the structural and mechanistical difference of the solubilization factors PrBP/δ and RhoGDI may arise from their different physiological roles as trafficking or regulatory proteins, respectively.

## Methods

**Preparation of isolated disc membranes.** Rod outer segments (ROS) were prepared from frozen bovine retinas[50]. Briefly, under dark conditions (red light) a 45% sucrose solution of phosphate buffer (67 mM K$_2$HPO$_4$/KH$_2$PO$_4$ (pH 7.0), 1 mM Mg(CH$_3$COO)$_2$ × 4H$_2$O, 100 μM EDTA-K$_2$, 1 mM DTT, 100 μM PMSF, 100 μM Aprotonin, 5 μM Leupeptin) containing 100 retinae was agitated for 2 min to break the outer segments. The detached ROS were filtered through a cotton net and subjected to discontinuous sucrose density-gradient (ρ = 1.11 g cm$^{-3}$, 1.13 g cm$^{-3}$, 1.15 g cm$^{-3}$) in phosphate buffer and centrifuged (15,000 × g, 20 min, 4 °C). The step-gradient contained ROS between the upper and the middle layer. The plasma membrane of ROS was ruptured open under osmotic conditions in buffer 20 mM 1,3-bis(tris(hydroxymethyl)methylamino)propane (BTP) (pH 7.0), 120 mM KCl, 0.2 mM MgCl$_2$, 5 mM DTT. Subsequent homogenization generated disc membranes. Isolated disc membranes were prepared from rod outer segments by two consecutive extractions using centrifugation (48,000 × g, 20 min, 4 °C)[51]. The pellet from the last centrifugation step was taken in isotonic buffer 20 mM BTP (pH 7.5), 130 mM NaCl, 1 mM MgCl$_2$ and 1 mM tris(2-carboxyethyl)phosphin (TCEP). Rhodopsin concentration was determined from its absorption spectrum using ε$_{500}$ = 40,000 M$^{-1}$ cm$^{-1}$.

**Purification of G protein.** Native G protein (transducin) was extracted from bovine rod outer segments[52]. Briefly, ROS from 100 retinae were re-suspended in isotonic buffer 20 mM BTP (pH 7.0), 120 mM KCl, 0.2 mM MgCl$_2$, 5 mM DTT at 4 mg rhodopsin mL$^{-1}$. Subsequently, they were homogenized, pelleted and re-suspended in hypotonic buffer 5 mM Tris–HCl (pH 7.0), 5 mM DTT at 1 mg rhodopsin mL$^{-1}$ without any detergents. Subsequent exposure to bright orange light (filter GG495) for 10 min resulted in a tight membrane–transducin complex, allowing removal of PDE6 in two consecutive centrifugal washing steps (90,000 × g, 30 min, 4 °C). The PDE6 free pellet was incubated with 150 μM GTP and 50 μM MgCl$_2$ for 10 min to allow dissociation of G-protein into its subunits and from the membranes. Gα and Gβγ subunits were separated on Blue-Sepharose column (1 mL HiTrap Blue, GE Healthcare, Germany) in buffer 20 mM BTP (pH 7.5), 1 mM MgCl$_2$, 2 mM DTT in a continuous NaCl gradient (0–300) for 25 mL followed by step gradient 1 M for another 25 mL[53]. Separated subunits were concentrated to 20 μM (centricon YM10, Millipore). GαGTPγS (Gα*) was prepared by activation of isolated Gα (20 μM) with twofold molar excess of GTPγS (10 min incubation at room temperature) in presence of 0.5 μM rhodopsin in isolated disc membranes. After removal of the membranes by centrifugation, isolated Gα* was stored at −40 °C.

**Purification of PDE6.** Native PDE6 was extracted from bovine rod outer segments (ROS) in hypotonic buffer without any detergents, as described[52]. Briefly, ROS from 100 retinae were re-suspended in isotonic buffer 20 mM BTP (pH 7.0), 120 mM KCl, 0.2 mM MgCl$_2$, 5 mM DTT at 4 mg rhodopsin mL$^{-1}$. Subsequently, they were homogenized, pelleted and re-suspended in hypotonic buffer 5 mM Tris-HCl (pH 7.0), 5 mM DTT at 1 mg rhodopsin mL$^{-1}$ without any detergents. Subsequent exposure to bright orange light (filter GG495) for 10 min resulted in tight membrane–transducin complex, so that PDE6 could be extracted in two consecutive centrifugal washing steps (90,000 × g, 30 min, 4 °C) in buffer 5 mM Tris–HCl (pH 7.0), 5 mM DTT. PDE6 was further purified by TSK-heparin column chromatography at a flow rate of 1 mL min$^{-1}$ in buffer 20 mM BTP (pH 7.5), 1 mM MgCl$_2$, 2 mM DTT, as described[54]. A salt gradient (0–600 mM NaCl) was produced over 30 mL. The homogenous PDE6 eluent was dialyzed against 20 mM 1,3-bis(tris(hydroxymethyl)methylamino)propane (BTP) (pH 7.5), 130 mM NaCl, 1 mM MgCl$_2$, and 1 mM tris(2-carboxyethyl)phosphin (TCEP) and concentrated to 10–20 μM (centricon YM30, Millipore). Purified PDE6 was stored at −40 °C in 20% glycerol. For electron microscopy, PDE6 (50 μl) was further purified prior to

the experiments by gel filtration (Äktamicro system, GE Healthcare, Germany) in 20 mM BTP (pH 7.5), 130 mM NaCl, 1 mM MgCl$_2$ and 1 mM TCEP at 4 °C using Superdex 200 GL 5/150 columns (GE Healthcare, Germany).

**Expression and purification of PrBP/δ.** N-terminally His-tagged PrBP/δ was cloned in the expression vector pET28b + (Novagen) from a template generously gifted by Prof. Dr. W. Baehr (University of Utah, USA). For amplification, NdeI based forward primer (5′-CTCCATATGTCAGCCAAGGACGAGC-3′) and NotI based reverse primer (5′-CTCGCGGCCGCTCAAACATAGAAAAG CCTCACTTTG-3′) were used. Protein expression was carried out as published for the original template[12]. Briefly, BL21-DE3 strain of E. coli was transformed with expression vector starter culture and grown overnight in 2YT media. The main culture was started by diluting the starter culture 1:100 in Autoinduction media by Novagen (37 °C, 200 rpm, 24 h). Cell lysis was carried out using 5 cycles of 30 s sonication each with 30 s of intermittent pause for cooling in PBS buffer without any detergents. His-tagged PrBP/δ was purified over a 1 mL NiTA column (GE Healthcare, Germany) in a gradient (0–500 mM) of imidazole in 50 mM Tris–HCl (pH 7.4), 500 mM NaCl and 1 mM dithiothreitol (DTT) buffer. The protein eluted at ~150 mM imidazole and was further purified by gel filtration in 20 mM BTP (pH 7.5), 130 mM NaCl, 1 mM MgCl$_2$, 1 mM TCEP at 4 °C. Though His-tag was tested not to influence the functional properties of the PrBP/δ, it was cleaved for activity experiments by incubation with thrombin protease, according to manufacturer's protocol (GE Healthcare, Germany). The purified PrBP/δ protein was stored at −40 °C.

**Size-exclusion chromatography of the PrBP/δ–PDE6 complex.** For determining the stoichiometry of the complex, PDE6 concentration of ~2 μM with molar ratios of 1, 2 and 3 for PrBP/δ were subjected to gel filtration (50 μl sample volume, Superose 12 column, GE Healthcare, Germany) using SMART systems (GE Healthcare, Germany) in 20 mM BTP (pH 7.5), 130 mM NaCl, 1 mM MgCl$_2$, 1 mM TCEP at 10 °C. The column was previously calibrated using external protein standards (Thyroglobin (669 kDa), Ferritin (440 kDa), Catalase (232 kDa), Alcohol Dehydrogenase (158 kDa), Albumin (67 kDa), Ovalbumin (43 kDa), Chymotrypsine (25 kDa), RNase (13.7 kDa)).

**Quantification of PDE6 membrane-association.** A centrifugal pull-down assay[55] was carried out to separate and quantify membrane-association of PDE6 in parallel to the activity measurements. 50 μl samples containing PDE6, PrBP/δ, Gα* and isolated disc membranes were subjected to centrifugation at 14,000 × g for 5 min at 22 °C. After removing the supernatant, the pellet was washed once with 50 μl buffer. Supernatant and pellet fractions were subjected to SDS–PAGE analysis. PDE6 was densitometrically quantified in the coomassie-stained gels (GelAnalyzer). The results depicted in Supplementary Fig. 5 show that on average (67.67 ± 12.17)% of PDE6 is membrane bound.

**PDE6 activity assay.** PDE6 catalyzed hydrolysis of cGMP generates GMP and a proton, whose evolution changes the pH of buffer. The rate of this change was monitored in real-time using a fast response micro pH electrode (Hach-Lange, Germany)[56]. Activity of PDE6 (50 nM) was measured in 20 mM BTP (pH 7.5), 130 mM NaCl, 1 mM MgCl$_2$, 1 mM TCEP (120 μl final volume) at 22 °C in presence of washed ROS membranes (10 μM), Gα* (1 μM) and varying concentrations of PrBP/δ. Measurements were initiated by addition of 2.5 mM cGMP and the change in pH of the sample was monitored with a dwell time of 50–200 ms. PDE6-activitiy was estimated from the initial slope of the pH-change. Titrations with known HCl/NaOH solutions at the end of the measurements were used to convert the pH change to concentration of cGMP hydrolyzed.

Under the conditions of the activity assay, membrane bound PDE6 (PDE6$_m$) is in equilibrium with soluble PDE6 (PDE6$_s$) with an apparent dissociation constant $K_1 = [\text{PDE}_m]\,[\text{PDE}_s]^{-1}$. A value of $K_1 = 2$ was obtained by quantifying $[\text{PDE}_m]$ and $[\text{PDE}_s]$ using the centrifugal pull-down assay (Supplementary Fig. 5). The enzymatic activity of PDE$_s$ ($358 \pm 28$ [cGMP][PDE$_s$]$^{-1}$ s$^{-1}$) was estimated in control experiments (50 nM PDE6, 1 μM Gα*) in the absence of membranes. The enzymatic activity of PDE$_m$ ($2755 \pm 69$ [cGMP][PDE$_m$]$^{-1}$ s$^{-1}$) was then calculated with $K_1$ and the overall PDE6 activity measured in the presence of membranes under otherwise identical conditions. The influence of PrBP/δ on PDE6 activity was analyzed using the reaction model depicted in Fig. 5b. According to the model, PDE6$_s$ comprises two independent PrBP/δ-binding sites potentially with different affinities for PrBP/δ (dissociation constants $K_2$ and $K_3$). It is already shown that interaction of PrBP/δ with PDE6s does not affect the enzymatic activity[11]. Since PrBP/δ binds exclusively to PDE6s the decrease of the overall Gα*-induced PDE6 activity results from the PrBP/δ induced solubilization of PDE6 (Fig. 4c). The data points of the PDE6 activity as a function of PrBP/δ were numerically fitted using Scientist software (MicroMath). In the fitting procedure $K_2$ and $K_3$ were allowed to vary with fixed values for $K_1$ and the enzymatic activities of PDE$_m$ and PDE$_s$.

**Crystallization of PrBP/δ.** Gel filtration-purified N-terminally His-tagged apo-PrBP/δ was concentrated to 12 mg mL$^{-1}$ in 20 mM BTP (pH 7.5), 130 mM NaCl, 1 mM MgCl$_2$, 1 mM TCEP at 4 °C. Crystals were grown using the hanging drop method. Crystallization was performed at 18 °C. The precipitant solution consisted

of 0.1 M Tris-HCl (pH 9.0), 0.1 M sodium acetate (pH 5.1), 8% polyethylene glycol (PEG) 8000 and 15% isopropanol. Each drop was prepared in a drop ratio 1:1. For cryo-protection crystals were soaked for several minutes in mother liquor containing 28% PEG 8000 and afterwards flash frozen in liquid nitrogen.

**Data collection and structure analysis.** Diffraction data collections of crystals were performed at 100 K using synchrotron X-ray sources at ESRF (Grenoble, France) and BESSY II (Berlin, Germany). The best diffraction data for the highest resolution of apo-PrBP/δ was collected at synchrotron beamline ID29 at ESRF (Grenoble, France) with a Pilatus 6 M detector at 0.966 Å wavelength[57]. All images were indexed, integrated and scaled using the XDS program package[58] and the CCP4 program SCALA[59–61]. Apo-PrBP/δ crystals belonged to the orthorhombic space group $P2_12_12_1$ (a = 27.66 Å, b = 56.53 Å, c = 89.10 Å, α = 90°, β = 90°, γ = 90°). Table 1 summarizes the statistics for crystallographic data collection and structural refinement. Initial phases for apo-PrBP/δ were obtained by the conventional molecular replacement protocol (rotation, translation, rigid-body fitting) using the crystal structure of human PrBP/δ crystallized in complex with RPGR RCC1-like domain (PDB entry 4JHP)[21] as the initial search model, which was used in the program PHASER[62]. A simulated annealing procedure with the resulting model was performed using a slow-cooling protocol and a maximum likelihood target function, energy minimization, and B-factor refinement by the program PHENIX[63], which was carried out in the resolution range of 47.74–1.81 Å for the apo-PrBP/δ structure. Apo-PrBP/δ was modeled with TLS refinement[64] using anisotropic temperature factors for all atoms. Restrained, individual B-factors were refined, and the crystal structure was finalized by the CCP4 program REFMAC5[65] and other programs of the CCP4 suite[59]. The final model has agreement factors $R_{\text{free}}$ and $R_{\text{cryst}}$ of 21.4 and 18.3%, for apo-PrBP/δ. Manual rebuilding of the apo-PrBP/δ model and electron density interpretation were performed after each refinement cycle using the program COOT[63,66]. We used additional simulated annealing as an optimization method performed with the PHENIX suite to remove model bias and displayed the results as "simulated annealing composite omit" maps of the apo-PrBP/δ cavity (Supplementary Fig. 1). Structure validation was performed with the programs PHENIX[63], SFCHECK[67], PROCHECK[68], WHAT_CHECK[69], and RAMPAGE[70]. Potential hydrogen bonds and van der Waals contacts were analyzed using the programs HBPLUS[71] and LIGPLOT[72]. All crystal structure superposition's of backbone α-carbon traces were performed using the CCP4 program LSQKAB[59]. All molecular graphics representations in this work were created using PyMol[73].

**Electron microscopy of the PDE6–PrBP/δ complex.** For formation of PDE6–PrBP/δ complex, purified PrBP/δ (0.5 μM final concentration) was mixed with purified full-length PDE6 (50 nM final concentration) prior to performing negative-stain electron microscopy. The samples were applied to freshly glow-discharged holey carbon grids (R2/4 Quantifoil grids, Quantifoil Micro Tools GmbH, Jena, Germany) covered with an additional thin carbon support film, and stained using uranyl acetate (2% w v$^{-1}$). Transmission electron microscopy images were collected on a Tecnai G2 Spirit microscope (FEI) operated at 120 kV, which was equipped with an Eagle 2k CCD camera (FEI). Micrographs were collected at ×42,000 magnification and a defocus range of −1 to −3 μm using the Leginon system[74]. The pixel-size at the object plane corresponds to 2.6 Å/pixel.

Defocus estimation was performed using CTFFIND[75], and micrographs were pre-selected based on calculated defocus and astigmatism values. This yielded a total of 92 good micrographs for PDE6 and 93 good micrographs for PDE6–PrBP/δ. In total 14634 particles images for PDE6 were identified using the Swarm tool in EMAN2[76] and manual intervention. Particle images were phase-flipped to correct for effects of the contrast transfer function, subjected to reference-free alignment and then classified using ISAC[77]. 280 classes of PDE6 particles were generated and 13 classes (702 particles) showed one or two additional protruding densities, interpreted to be ~17 kDa endogenous PrBP/δ. Similarly, 14963 particles for PDE6–PrBP/δ and classified using ISAC. 253 classes of PDE6–PrBP/δ were generated and 188 classes (11648 particles) showed one (PDE6–PrBP/δ complex) or two (PDE6–PrBP/δ$_2$ complex) additional protruding densities, interpreted to be ~17 kDa PrBP/δ added to sample.

**Data availability.** The atomic coordinates and structure factors have been deposited in the Protein Data Bank under accession code 5T4X. Other data are available from the corresponding author upon reasonable request.

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

## Acknowledgements

We thank Prof. Dr. Wolfgang Baehr (University of Utah, USA) for providing us PrBP/δ expression vector. We are obliged to Ms. Ingrid Semjonow, Ms. Helena Seibel and Mr. Brian Bauer for their excellent technical assistance in purifying reagents and in protein preparations. We are grateful to Uwe Müller, Manfred Weiss and the scientific staff of the BESSY-MX/Helmholtz Zentrum Berlin für Materialien und Energie at beamlines BL14.1, BL14.2 and BL14.3 operated by the Joint Berlin MX-Laboratory at the BESSY II electron storage ring (Berlin-Adlershof, Germany) and the scientific staff of the European Synchrotron Radiation Facility (ESRF, Grenoble) at beamlines ID14-1, ID14-4, ID23-1, ID23-2, ID30A, ID30B and ID29, where the data were collected, for continuous support. This work was supported by grants from the Deutsche Forschungsgemeinschaft (SFB740 to T.M., C.M.T.S, M.H., P.S.; SFB1078-B6 to P.S.; SFB958 to C.M.T.S.), DFG Cluster of Excellence 'Unifying Concepts in Catalysis' (Research Field D/E to P.S.). E.B. holds a Freigeist-Fellowship from the Volkswagen Foundation and acknowledges continuous support from the Caesar Foundation.

## Author contributions

M.H. and P.S. designed the research; B.M.Q. and A.S. performed protein purification; B.M.Q performed membrane purifications, obtained enzyme kinetics; B.M.Q, J.B and T.M. collected electron microscopy data. A.S. and P.S. crystallized the protein and obtained structural data; A.S. optimized the crystallization procedure; A.S. and P.S. carried out protein X-ray crystallography; E.B. and T.M. carried out structural analysis of the electron microscopy data. T.M. and C.M.T.S. supervised electron microscopy data, M.H. analyzed enzyme kinetics data. B.M.Q., A.S., E.B., M.H., and P.S. wrote the manuscript with contributions from all co-authors; all authors reviewed the results and approved the final version of the manuscript.

## Additional information

**Competing interests:** The authors declare no competing financial interests.

