## [Peer Review File · Nature Communications]

REVIEWER COMMENTS:

Reviewer #1:

In the manuscript by Bilal et al, the authors report the apo structure of PDE6D and a low resolution EM structure of a PDE6/PDE6D complex. The authors propose that PDE6D does not extract prenylated proteins from the membrane but rather scavenge dissociated lipid modified PDE6

The manuscript is well written, however the model that authors propose has actually been shown before for PDE6D and published in Cell by Malta et al 2014 (paper title: KRas Localizes to the Plasma Membrane by Spatial Cycles of Solubilization, Trapping and Vesicular Transport) and I quote one of the questions in that published paper "In order to understand how KRas solubilization can contribute to PM enrichment, we investigated whether PDE δ extracts KRas directly from membranes or whether it passively sequesters a cytosolic fraction of KRas that is in equilibrium with membranes" and their conclusion was "PDE δ Accelerates Equilibration by Passively Sequestering Soluble KRas"

Furthermore although the conclusion is correct (which was published before) the experiment of membrane fractionation the authors show does not prove passive removal of PDE6 from membranes. In order to do that the authors need to perform a kinetic assay instead

Based on that I don't think these findings are of great significance to be published in Nature communications

Reviewer #2:

This manuscript presents the structure of an unoccupied prenyl binding protein (PrBP/ δ). In comparison with multiple previous structures of this protein in occupied form, it seems that Apo-PrBP/ δ harbors a pre-shaped cavity that permits binding of prenylated cargo. The authors show data that suggest that a farnesylated and geranylgeranylated protein, PDE6, binds two moles PrBP. They propose that equilibrium dissociation from membranes coupled to PrBP binding provides a mechanism for transfer of prenylated substrates between membranes, distinct from RhoGDI. The work is of high quality and deserves presentation in a good journal after addressing the following points.

1. Is it not possible that the protein could also exist in a closed conformation that was not crystallized? Details of the protein purification are not provided to ascertain if a hydrophobic constituent might have been present (nor is there any mention of how prenylated proteins were kept soluble later in the ms.). Please discuss and write the conclusions stating that the structure SUPPORTS a model in which an open conformation predominates....
2. Page 3. The authors state that GTP hydrolysis provides the energy to drive the trafficking cycle. This is not correct. It would be driven by binding energy of a GTP bound form of the GTP binding protein, stabilizing distinct conformations.
3. The text is missing some language articles (the, etc) and will benefit from minor editing

Reviewer#3:

The manuscript of Qureshi et al. presents the first structure of prenyl binding protein (PrBP/delta) in the absence of ligands. The authors present evidence that isolated PDE6 binds to their recombinant PrBP/delta that was used for the structure determination. Their results are consistent with one PDE6 binding to two PrBPdelta molecules, one on the farnesyl moiety and one on the geranyl moiety. The authors propose that that PrBP/delta functions by binding to PDE6 in solution only, and that it has no ability to extract the PDE6 from membranes.

This is a very well written manuscript that does an admirable job of clearly placing the current results in the context of the extensive structural and functional work that has been carried out for PrBP/delta, UNC119 and RhoGDI. It is a real pleasure to read the manuscript. For Nature Communications, I would expect that there should be significant novel aspects to the work. It seems to me that the three novel points made by the manuscript are:

1. This is the first apo PrBP/delta structure. There are already lots of structures with ligands, but this is the first apo structure.
2. The hydrophobic cavity in the beta-sandwich is preformed, even though the cavity is empty. This is a surprising and novel finding.
3. The mechanism of solubilization by PrBP/delta is passive and simply functions by mass action. The authors propose that PrBP/delta binds only to soluble PDE6. It has no role in actively extracting PDE6 from membranes.

I believe the combination of the above points really would constitute a valuable contribution to Nature Communications. My concern is that the authors have not conclusively demonstrated the above points.

1. This is the first apo structure of PrBP/delta. However, the authors have not clearly stated in the manuscript that there is no density in the pocket. Is there any indication that the pocket may have picked up lipid from the bacteria? Is there weak density here? The crystallization condition includes 15% isopropanol. Is there indication of bound isopropanol? Is the conformation in the crystals truly the same as the conformation that it would have in an aqueous solution devoid of isopropanol? Does hydrogen/deuterium exchange suggest that isopropanol has no effect on the residues in the cavity (NMR or mass spec measurements)? Does isopropanol affect the interaction of the protein with lipids?

2. The hydrophobic cavity is preformed. This appears to be the case in 15% isopropanol. Is it the case for a more physiological aqueous solution?

3. If the mechanism that the authors propose for the PrBP/delta-mediated solubilization, then initial rates of dissociation of PDE6 from membranes should not be affected by the presence of the PrBP/delta. I can appreciate that measurement of these initial rates may not be straightforward. It would likely involve stopped-flow experiments with labeled PDE6, or single-molecule measurements with TIRF illumination.

Of these three points, the first two are of the greatest concern.

There are a few minor points that should be corrected:

1. The resolution range was quoted as 47.71-1.81 in Supp Table 1. However in parentheses, it says

highest resolution shell is: (2.90-2.75). This appears to be cut and paste damage that should be corrected.

2. In sup Table 1, cc^* should be $cc^{1/2}$, using Nature's format. Also nature format says B factors for proteins, ligands and water should be listed separately.

3. How was the single Na^+ identified?

4. p. 5 line 110 "2Fo-Fc" should probably be "2mFo-DFC", i.e., the σ_A weighted map. If not, the authors should explain in the methods why they did not use the σ_A weighting.

5. p. 9, line 195. The statement

6. p. 9, line 200,
"Beside Ar12" should be "Besides Ar12"

7. The legend for Fig. 4 describes the colors of the traces, but this should be indicated on the figure as titles connected to the correct curve by a thin line. This will help color-blind readers.

8. In the legend for Figure 4b, the text states that the complex has one or two additional densities that presumably correspond to the PrBP/delta. The authors should include a few arrows so that it is unambiguous as to what density they refer.

9. On p. 11, line 261, the authors state:

"the data points of the titration curve shown in Fig. 4c are well fitted with two KD values". Actually, the points appear to be equally well fitted by a single type of site, so this statement is misleading.

R^2 is 0.996 for two sites and 0.994 for one site. If this is statistically significant, the authors should describe how they arrive at this significance.

To revision on manuscript NCOMMS-17-17087

Point by Point letter as response to the Reviewer:

We thank the referees for their thoughtful and constructive comments. We have addressed your concerns below, and have incorporated most of your suggestions in the revised manuscript. Your comments are in black text and our responses are in blue text.

Reviewers' comments

Reviewer #1 (Remarks to the Author):

In the manuscript by Bilal et al, the authors report the apo structure of PDE6D and a low resolution EM structure of a PDE6/PDE6D complex. The authors propose that PDE6D does not extract prenylated proteins from the membrane but rather scavenge dissociated lipid modified PDE6

The manuscript is well written, however the model that authors propose has actually been shown before for PDE6D and published in Cell by Malta et al 2014 (paper title: KRas Localizes to the Plasma Membrane by Spatial Cycles of Solubilization, Trapping and Vesicular Transport) and I quote one of the questions in that published paper "In order to understand how KRas solubilization can contribute to PM enrichment, we investigated whether PDE δ extracts KRas directly from membranes or whether it passively sequesters a cytosolic fraction of KRas that is in equilibrium with membranes" and their conclusion was "PDE δ Accelerates Equilibration by Passively Sequestering Soluble KRas"

Response

First of all, we would like to thank the reviewer for pointing out the study by Schmick *et al.* (2014), which indeed comes to similar conclusion and we have now discussed it in our revised manuscript.

Schmick *et al.* employ cell-based *in vivo* assays to conclude that PrBP/ δ solubilizes KRas by sequestering soluble species. In strong contrast, we used prenylated PDE6 as a binding partner and performed *in vitro* experiments with purified components, which allow new titration experiments and thereby yield K_D values of the interactions. Furthermore, our structural data strongly support the proposed model and now allow a molecular interpretation of the solubilization mechanism. We thus believe that our study adds very important novel aspects regarding the solubilization mechanism of prenylated-cargo by PrBP/ δ and that due to the different focus, methods und binding partner used the two investigations support each other.

We modified the abstract and discussion accordingly:

(page 2, lines 31-33): "To investigate the molecular mechanism of cargo solubilization we analyzed the PrBP/ δ -induced membrane dissociation of rod photoreceptor phosphodiesterase (PDE6)."

(page 13, lines 297-308): "Several lines of evidence suggest that the mechanism of cargo extraction from membranes differs between RhoGDI and PrBP/ δ . Recently, Schmick and colleagues have performed cell-based *in vivo* investigations on PrBP/ δ mediated solubilizaion of Kras from endo-membranes and concluded that binding of KRas to PrBP/ δ is a passive sequestration of spontaneously dissociated KRas⁴⁰. Consistently, there is no evidence for formation of a membrane-bound complex of PrBP/ δ with cargo-proteins¹⁵. From a structural point of view PrBP/ δ lacks an N-terminal domain that could function like the regulatory arm of RhoGDI. It is thus unlikely that PrBP/ δ can interact with its prenylated binding partner prior to its dissociation from the membrane. In our crystal structure, isolated PrBP/ δ (apo- PrBP/ δ) exhibits an open conformation, *i.e.* harbors a deep, hydrophobic cavity even without accommodating a prenyl-moiety. Taken together, these findings strongly support a "dissociation by depletion" mechanism for PrBP/ δ (Fig. 5)."

Furthermore although the conclusion is correct (which was published before) the experiment of membrane fractionation the authors show does not prove passive removal of PDE6 from membranes. In order to do that the authors need to perform a kinetic assay instead. Based on that I don't think these findings are of great significance to be published in Nature communications.

Response

Our investigation focuses on a molecular and structural interpretation of the solubilization mechanism of prenylated proteins by PrBP/ δ . Since PrBP/ δ lacks an N-terminal domain that could function like the regulatory arm of RhoGDI it is likely that the mechanism of extraction of membrane associated cargo-proteins by PrBP/ δ and RhoGDI, respectively, is different. The here proposed mechanism is based on the novel finding of a ligand-free, preshaped hydrophobic cavity of apo-PrBP/ δ . The titration experiments were primarily performed to quantify the affinity of PrBP/ δ for PDE6. We agree with the reviewer that these measurements - although consistent with the proposed mechanism - do not unequivocally prove the passive removal of PDE6 from membranes, which we have clearly stated in the manuscript. We note that cell-based *in vivo* experiments as performed by Schmick *et al.* (2014) cannot be performed because recombinant expression of functionally intact, labeled PDE6 is to date not possible.

A detailed kinetic analysis of PDE6 solubilization by labeled PrBP/ δ will be an important part of further investigations but would go beyond the scope of our present study.

Reviewer #2 (Remarks to the Author):

This manuscript presents the structure of an unoccupied prenyl binding protein (PrBP/ δ). In comparison with multiple previous structures of this protein in occupied form, it seems that Apo-PrBP/ δ harbors a preshaped cavity that permits binding of prenylated cargo. The authors show data that suggest that a farnesylated and geranylgeranylated protein, PDE6, binds two moles PrBP. They propose that equilibrium dissociation from membranes coupled to PrBP binding provides a mechanism for transfer of prenylated substrates between membranes, distinct from RhoGDI. The work is of high quality and deserves presentation in a good journal after addressing the following points.

1. Is it not possible that the protein could also exist in a closed conformation that was not crystallized? Details of the protein purification are not provided to ascertain if a hydrophobic constituent might have been present (nor is there any mention of how prenylated proteins were kept soluble later in the ms.). Please discuss and write the conclusions stating that the structure SUPPORTS a model in which an open conformation predominates....

Response

We added more details of protein purification to the manuscript (page 17). We have not used any detergents to solubilize or to stabilize the protein during purification. PrBP/ δ is very soluble up to at least 50 mg/ml concentration. Moreover, the prenylated-binding partner PDE6 is also soluble without lipids or detergents once extracted from membranes, and the extraction is performed by washing with hypotonic buffer without any detergents. PDE6 remains soluble up to 50 mg/ml concentration in buffer without any detergents or membranes. We could not observe any continuous electron density in the empty binding cavity of PrBP/ δ (added in a new supplementary figure 1) which support the conclusion that no detergents or other hydrophobic molecules from the buffer were picked up by the protein during purification or crystallization (for more details regarding the occupancy of the binding pocket please see response to reviewer 3).

Support for a model in which the open PrBP/ δ conformation predominates in the absence of cargo comes from the structure of the RPGR-PrBP/ δ complex (Wätzlich *et al.*, 2013). In this structure PrBP/ δ is cargo-free and in

a similar open conformation. Furthermore, if the closed conformation of Apo-PrBP/ δ would predominate in solution, a cargo-induced opening of the binding pocket is needed for efficient and fast cargo solubilization, i.e. a mechanism similar to RhoGDI. As pointed out above and in the manuscript, an “induced fit” mechanism of PrBP/ δ -cargo binding is unlikely because PrBP/ δ lacks the N-terminal domain that could function like the regulatory arm of RhoGDI. An experimental proof that apo-PrBP/ δ is in the open conformation in solution (e.g. by site-directed labeling and EPR or advanced fluorescence measurements) will be an important part of further investigations but would go beyond the scope of our present study.

2. Page 3. The authors state that GTP hydrolysis provides the energy to drive the trafficking cycle. This is not correct. It would be driven by binding energy of a GTP bound form of the GTP binding protein, stabilizing distinct conformations.

Response

Here we respectfully disagree: binding energies provide the energy for individual steps within a reaction sequence. In a run through an entire reaction cycle (i.e. initial and final state of the reactants are identical), however, binding and dissociation energies eventually cancel each other. Hence an additional energy input such as ATP or GTP hydrolysis is generally required to drive reaction cycles (note that we do not address the question how this energy is used in individual steps of the cycle). To make our statement more clearly, we modified the text accordingly:

(page 5, line 98-99): "...hydrolysis of GTP provides the net-energy input to run through the entire PrBP/ δ trafficking cycle."

3. The text is missing some language articles (the, etc) and will benefit from minor editing

Response

We modified the text accordingly.

Reviewer #3 (Remarks to the Author):

The manuscript of Qureshi et al. presents the first structure of prenyl binding protein (PrBP/ δ) in the absence of ligands. The authors present evidence that isolated PDE6 binds to their recombinant PrBP/ δ that was used for the structure determination. Their results are consistent with one PDE6 binding to two PrBP/ δ molecules, one on the farnesyl moiety and one on the geranyl moiety. The authors propose that PrBP/ δ functions by binding to PDE6 in solution only, and that it has no ability to extract the PDE6 from membranes.

This is a very well written manuscript that does an admirable job of clearly placing the current results in the context of the extensive structural and functional work that has been carried out for PrBP/ δ , UNC119 and RhoGDI. It is a real pleasure to read the manuscript. For Nature Communications, I would expect that there should be significant novel aspects to the work. It seems to me that the three novel points made by the manuscript are:

1. This is the first apo PrBP/ δ structure. There are already lots of structures with ligands, but this is the first apo structure.
2. The hydrophobic cavity in the beta-sandwich is preformed, even though the cavity is empty. This is a surprising and novel finding.
3. The mechanism of solubilization by PrBP/ δ is passive and simply functions by mass action. The authors

propose that PrBP/delta binds only to soluble PDE6. It has no role in actively extracting PDE6 from membranes.

I believe the combination of the above points really would constitute a valuable contribution to Nature Communications.

My concern is that the authors have not conclusively demonstrated the above points.

1. This is the first apo structure of PrBP/delta. However, the authors have not clearly stated in the manuscript that there is no density in the pocket. Is there any indication that the pocket may have picked up lipid from the bacteria? Is there weak density here? The crystallization condition includes 15% isopropanol. Is there indication of bound isopropanol? Is the conformation in the crystals truly the same as the conformation that it would have in an aqueous solution devoid of isopropanol? Does hydrogen/deuterium exchange suggest that isopropanol has no effect on the residues in the cavity (NMR or mass spec measurements)? Does isopropanol affect the interaction of the protein with lipids??

Response

We agree that this point should be discussed in the revised paper. As we noted above in our response to Reviewer 2, we have now added a new supplementary figure 1 with a calculated omitted and simulated annealing omit electron density maps of the binding cavity (Figure 1 below: new “Supplementary figure 1” in manuscript). The nearly unbiased simulated-annealing omit map was calculated for the cavity amino acids with starting simulation temperature of 5,000 K. There is no indication for any kind of continuous electron density in the binding cavity, *i.e.* no lipid or detergent is specifically bound. Therefore, we conclude that the protein has not picked up lipids during expression and purification. However, a small electron density indicates the presence of a highly flexible and presumably small molecule in the binding cavity. Since the density is located close to Arg61 we assume a polar molecule in H-bond distance. We thus recalculated the structure coordinates of PrBP/ δ and added two additional water molecules at this position (Figure 1: new “Supplementary figure 1”). In general, any larger cavities in proteins are *per se* not empty but often filled up with disordered water or solvent molecules.

Figure 1: New Supplementary Figure 1 | 2mFo-DFc omit map of the apo-PrBP/ δ binding cavity

The new structure coordinates of apo-PrBP/ δ with this two additional water molecules are uploaded in the PDB structure database. Supplementary table 1, all crystal structure figures (Fig.2 and 3 + Supplementary figures 1-3) and RMSD values in the manuscript are modified accordingly.

As stated above we modified the manuscript (page 6, line 118-126): “As mentioned in the introduction, two basic conformations of PrBP/ δ exist, “open” with a deep cavity and “closed” with a reduced cavity size. As we compare and discuss in greater detail in the following paragraph, we obtained the open conformation of PrBP/ δ . In this the apo-state a preformed cavity may allow an unspecific exchange of small molecules. We found in the cavity some, but not well-defined, electron density which indicate the presence of a flexible, small and polar molecule. Therefore we assigned this electron density as water molecules, which are located in H-bond distance to R61. (Supplementary Fig. 1).”

We refined the structure with isopropanol (IPA in the figure 2) bound in the cavity. This density maps clearly indicate that an isopropanol molecule does not fit adequately to the small electron density in the binding cavity.

Figure 2: 2mFo-DFc and difference mFo-Fc electron density maps of the apo-PrBP/ δ binding cavity with a refined isopropanol (IPA) in the calculation. The electron density does not fit proper to an IPA molecule.

Furthermore, we do not observe any electron density for isopropanol molecules bound to PrBP/ δ (in the cavity or at any other part of PrBP/ δ). Moreover, the high RMSD score between our structure and other crystal structures of PrBP/ δ in an open conformation obtained in the absence of isopropanol gives us the confidence that the structure represents the native open conformation for apo-PrBP/ δ . See also the answers to point 2 below.

2. The hydrophobic cavity is preformed. This appears to be the case in 15% isopropanol. Is it the case for a more physiological aqueous solution?

Response

We obtained similar structures from crystals grown in 6-10% isopropanol but with less resolution in the datasets ($> 2.5 \text{ \AA}$). We do not observe any ordered electron density for isopropanol molecules bound to PrBP/ δ in the cavity or at any other part of the PrBP/ δ . As an example we calculated omitted maps with bound isopropanol (IPA in the figure 2) in the binding cavity (figure 2). This density maps clearly indicate that an isopropanol molecule fit not adequate to the small and not well-ordered electron density in the binding cavity.

See also the answers to point 1 above.

3. If the mechanism that the authors propose for the PrBP/ δ -mediated solubilization, then initial rates of dissociation of PDE6 from membranes should not be affected by the presence of the PrBP/ δ . I can appreciate that measurement of these initial rates may not be straightforward. It would likely involve stopped-flow experiments with labeled PDE6, or single-molecule measurements with TIRF illumination.

Response

We thank the reviewer for these suggestions; the dependence of the initial rates of PDE6 dissociation on PrBP/ δ concentration would indeed prove the mechanism. As pointed out above (see answer to reviewer 1) cell-based experiments cannot be applied, because recombinant expression of functionally intact full-length PDE6 is not possible. Labeling of PDE6 for *in vitro* measurements is challenging and would involve expression and labeling of single cysteine PDE γ subunits and subsequent PDE γ subunit exchange in native PDE6. Kinetic characterization of PDE6 solubilization by PrBP/ δ will thus be part of further studies with other collaboration partner specialized on single-molecule or stopped-flow techniques, but would go beyond the scope of our present study.

There are a few minor points that should be corrected:

Thank you for bringing these points to our attention.

1. The resolution range was quoted as 47.71-1.81 in Supp.Table 1. However in parentheses, it says highest resolution shell is: (2.90-2.75). This appears to be cut and paste error that should be corrected.

Response

We modified the supplementary table 1 accordingly.

2. In sup Table 1, cc^* should be $cc1/2$, using Nature's format. Also nature format says B factors for proteins, ligands and water should be listed separately

Response

Per the reviewer's suggestion, we changed cc^* to $cc1/2$ according to the Nature's format. We expanded supplementary table 1 with B factors for proteins, ligands and water accordingly. Supplementary table1 has been revised as recommended.

3. How was the single Na^+ identified?

Response

As mentioned before, we re-calculated the data and in the new crystal structure we identified this density as a highly ordered water molecule instead of the initially proposed sodium. The coordination sphere for ion or water around this binding site is not obvious.

4. p. 5 line 110 “2Fo-Fc” should probably be “2mFo-DFC”, i.e., the σ_A weighted map. If not, the authors should explain in the methods why they did not use the sigma weighting.

Response

We renamed the “2Fo-Fc” to “2mFo-DFc” that is σ_A -weighted map.

5. p. 9, line 195. The statement “but points as other romater to the opposite direction” should be “points as another rotamer in the opposite direction”

Response

We modified the text accordingly.

6. p. 9, line 200,
“Beside Ar12” should be “Besides Ar12”

Response

We modified the text accordingly.

7. The legend for Fig. 4 describes the colors of the traces, but this should be indicated on the figure as titles connected to the correct curve by a thin line. This will help color-blind readers.

Response

We modified the Fig accordingly.

8. In the legend for Figure 4b, the text states that the complex has one or two additional densities that presumably correspond to the PrBP/delta. The authors should include a few arrows so that it is unambiguous as to what density they refer.

Response

We modified the Fig accordingly.

9. On p. 11, line 261, the authors state:
“the data points of the titration curve shown in Fig. 4c are well fitted with two KD values”. Actually, the points appear to be equally well fitted by a single type of site, so this statement is misleading. R^2 is 0.996 for two sites and 0.994 for one site. If this is statistically significant, the authors should describe how they arrive at this significance.

Response

The difference between the R^2 values of the two fits is indeed not significant, *i.e.* both models fit the data well. We thus deleted in the text "resulted in a poorer fit" and state now:

(page12, line 275): "Fitting of the titration data with a single type of binding sites on PDE6 for PrBP/ δ yields $K_D = 0.07 \pm 0.01 \mu\text{M}$ (see Fig. 4c)."

Despite the similar quality of the fits, we favor the model with two different binding sites because "the ~20-fold relative difference of the two values is in good agreement with previous results obtained with PrBP/ δ and C15-farnesyl-cysteine ($K_D = 0.7 \mu\text{M}$) and C20-geranylgeranyl-cysteine ($K_D = 19 \mu\text{M}$), respectively". Notably, as also stated in the manuscript, the proposed model of membrane extraction of PDE6 by PrBP/ δ is independent of whether the two PrBP/ δ binding sites on PDE6 are identical or not.